# Insulin and Its Key Role for Mitochondrial Function/Dysfunction and Quality Control: A Shared Link between Dysmetabolism and Neurodegeneration

**DOI:** 10.3390/biology11060943

**Published:** 2022-06-20

**Authors:** Giacoma Galizzi, Marta Di Carlo

**Affiliations:** Istituto per la Ricerca e l’Innovazione Biomedica (IRIB) CNR, Via Ugo la Malfa, 153, 90146 Palermo, Italy; giacoma.galizzi@irib.cnr.it

**Keywords:** insulin, insulin resistance, mitochondrial dysfunction, mitophagy, mitochondrial biogenesis

## Abstract

**Simple Summary:**

This review will comprise an overview of insulin and its history, structure, synthesis, secretion, signaling, and peripheral and brain roles. Further, the impact that metabolic diseases and the insulin resistance (IR) condition can exert on brain function will be surveyed. Later, attention will be given to the complex relation of mitochondrial dysfunction, brain insulin resistance, and neurodegenerative disease, especially Alzheimer’s disease (AD). We will also focus our consideration on the role played by insulin or the IR condition in mitochondrial homeostasis by analyzing the different processes involved in the quality control of mitochondria (MQC), such as mitochondrial dynamics, mitochondrial biogenesis, and selective autophagy (mitophagy), as well as pathophysiological implications of these aspects. Finally, the possibility of employing the mitochondrion as a target to fight dysmetabolism and AD-related effects will also be reviewed.

**Abstract:**

Insulin was discovered and isolated from the beta cells of pancreatic islets of dogs and is associated with the regulation of peripheral glucose homeostasis. Insulin produced in the brain is related to synaptic plasticity and memory. Defective insulin signaling plays a role in brain dysfunction, such as neurodegenerative disease. Growing evidence suggests a link between metabolic disorders, such as diabetes and obesity, and neurodegenerative diseases, especially Alzheimer’s disease (AD). This association is due to a common state of insulin resistance (IR) and mitochondrial dysfunction. This review takes a journey into the past to summarize what was known about the physiological and pathological role of insulin in peripheral tissues and the brain. Then, it will land in the present to analyze the insulin role on mitochondrial health and the effects on insulin resistance and neurodegenerative diseases that are IR-dependent. Specifically, we will focus our attention on the quality control of mitochondria (MQC), such as mitochondrial dynamics, mitochondrial biogenesis, and selective autophagy (mitophagy), in healthy and altered cases. Finally, this review will be projected toward the future by examining the most promising treatments that target the mitochondria to cure neurodegenerative diseases associated with metabolic disorders.

## 1. Introduction

Insulin is a hormone released from the pancreas and classically considered a hypoglycemic agent associated with controlling peripheral glucose homeostasis. Pancreatic beta cells are sensitive to blood sugar levels and operate as a switch, secreting insulin into the blood when glucose levels are high and reducing insulin secretion when glucose is low. In addition, insulin regulates the metabolism of carbohydrates, lipids, and proteins. Although insulin was often exclusively associated with a peripheral action, several studies revealed that insulin can also play a role in the brain since it regulates satiety in the hypothalamus. An abundant presence of insulin receptors has been identified in various brain regions [1]. These findings support the importance of insulin for brain physiological and vital functions, including energy maintenance, mood, and memory formation [2,3]. By binding to its receptor, insulin activates signaling whose defect plays a role in brain dysfunction, leading to neurodegenerative diseases, especially Alzheimer’s disease (AD) [4]. AD is the most common form of dementia, affecting millions of people worldwide, and it dramatically impacts the quality of life of both patients and caregivers. Since insulin signaling is essential for synaptic plasticity, memory, and long-term potentiation, the impaired insulin pathway is, consequently, a player in AD pathogenesis [5]. The link between insulin resistance and AD was widely demonstrated in rodent AD models, high-fat-diet-induced AD, and in a non-human primate (NHP) model [6,7]. In agreement with these results, clinical studies have correlated hyperinsulinemia with an increased risk of cognitive decline, showing an association between peripheral and brain insulin resistance in AD [8,9]. Moreover, insulin resistance and mitochondrial dysfunction are common features in AD, type 2 diabetes mellitus (T2D), and obesity, but what is the cause or the consequence is not entirely elucidated [10].

Mitochondrial dysfunction comprises impairment in mitochondrial bioenergetic dynamics and biogenesis, and it is closely connected with the alteration of insulin signaling [11]. 

For a long time, mitochondria were considered organelles separately fluctuating in the cytoplasm; successively, with the advent of new microscopic technologies, it was observed that mitochondria continually divide and fuse. The balance between fission and fusion in mitochondrial dynamics is important for growth redistribution and maintenance in a healthy mitochondria network and plays a role in disease-related processes, such as apoptosis and mitophagy [12]. All the cells consume energy for their homeostasis and specific activity, such as secretion, cytokines production, and neurotransmission, and require the support of functional mitochondria that provide ATP obtained by oxidative phosphorylation. A reduction in mitochondria respiration and bioenergetics is associated with insulin resistance [13]. Maintaining an appropriate number of mitochondria and related mtDNA is important for their functional state. Less and smaller-sized mitochondria are found in the skeletal muscle of insulin-resistant, obese, or T2DM subjects, suggesting biogenesis dysfunction [14]. 

Further, mitochondrial bioenergetic dynamics and biogenesis are joined to mitophagy, the process necessary to eliminate damaged organelles and maintain a stable number of healthy mitochondria within cells. Mitochondrial turnover is an aspect of quality control in which dysfunctional mitochondria are selectively eliminated through autophagy (mitophagy) and replaced through the extension of preexistent mitochondria (biogenesis). Dysfunction in these well-balanced mechanisms can contribute to dysmetabolic and neurodegenerative diseases. The review summarizes the effect/defect of insulin signaling in different features of mitochondrial dysfunction, focusing on dynamics, biogenesis, and mitophagy and their role in pathologies in which metabolic dysmetabolism is a comorbidity with neurodegeneration. 

## 2. Insulin Tales

In 1889, two German researchers, Oskar Minkowski and Joseph von Mering, hypothesized that a substance produced and secreted by the pancreas was responsible for glucose metabolic control [15]. Successively, it was noted that diabetes was associated with the destruction of the islets of Langerhans, discovering the place where the secretion was produced. 

The story tells that the success of insulin isolation was due to the collaboration of four different scientists. Frederick Grant Banting, an orthopedic surgeon, had the idea to isolate the pancreatic islets of Langerhans extracts by ligating the pancreatic duct of dogs. With the aim to test this hypothesis, he contacted John James Macleod, an expert in the diabetes field at the University of Toronto. Macleod ensured him laboratory facilities, ten dogs for his experiments, Charles Best, a young biochemistry student, as a research assistant, and provided supervision and mentoring. The experimental studies began in 1921 and showed that the depancreatized dog developed diabetes and that the intravenous injection of the pancreatic extract, called isletin, reduced the blood glucose [16]. However, after administration, the canine extract induced some toxic reactions. In 1922, James Collip developed a method to produce a non-toxic isletin that was sufficiently pure for human use. Then, the active substance was renamed “insulin”. On October 25, 1923, Banting and Macleod received the Nobel Prize in Physiology or Medicine for the discovery of insulin. However, Banting decided to share his prize money with Best, who helped him with the experiments, and MacLeod shared his money award with Collip, who purified the extract [17].

In 1923, August Krogh, whose wife had diabetes mellitus, asked for the authorization from the University of Toronto to bring insulin to Scandinavia and founded the nonprofit Nordisk Insulin Laboratory, beginning the production of insulin to be placed on the market [18]. In 1978, with the advent of the recombinant DNA (rDNA) technologies, David Goeddel and his colleagues of Genentech cloned human insulin by utilizing and combining the insulin A- and B-chains to be expressed in Escherichia coli. After that, Genentech and Lilly signed an agreement to commercialize rDNA insulin. 

Further Nobel Prizes were awarded to studies on insulin. In 1958, British molecular biologist Frederick Sanger obtained a Nobel Prize for his work on the structure of proteins, including insulin, the first protein whose sequence was determined. Then, American medical physicist Rosalyn Yalow obtained a Nobel Prize for developing a radioimmunoassay (RIA) to measure the concentration of insulin and other hormones in the blood. The same year, Roger Guillemin and Andrew V. Schally won the Nobel Prize for their discoveries concerning brain insulin production [17]. In 2021, the 100th anniversary of the discovery of insulin was celebrated.

## 3. Insulin Synthesis and Secretion

Nowadays, the insulin role is well-known. Specifically, it acts as a hormone secreted by the β cells of the pancreatic islets of Langerhans, whose main function is maintaining normal blood glucose levels by allowing cellular glucose uptake and by stimulating translocation of the glucose transporter GLUT-4 from intracellular sites to the plasma membrane. Healthy individuals are highly responsive to insulin in skeletal muscle, liver, and adipose tissue, whereas obese or type 2 diabetes mellitus (T2DM) individuals do not exert any reaction to the presence of insulin and, for this reason, are called insulin-resistant [19].

Further, it regulates cell metabolism and promotes cell division and growth through mitogenic activity. Contrariwise, insulin inhibits catabolic processes, such as gluconeogenesis, glycolysis, lipolysis, and proteolysis. 

Insulin is a post-translation product of a single-chain precursor named proinsulin. Its synthesis occurs on membrane-associated ribosomes present on the rough endoplasmic reticulum (RER), where it is folded and its disulfide bonds shaped. Secretory vesicles transport proinsulin from the ER to the Golgi apparatus, where a series of proteases process it to form mature insulin [20]. When mature granules are secreted into the circulation by exocytosis, insulin is released. Different can be the factors that stimulate insulin secretion, including modulation of gene transcription, translation, and post-translation in the Golgi apparatus [21]. Glucose is, of course, the main stimulus for insulin secretion, but other elements, such as nutrients, hormones, amino acids, and neuronal stimuli, may influence and modify this response [21]. Insulin secretion into the portal veins is a multioscillatory process due to the summation of synchronized secretion bursts from millions of β-cells in Langerhans islets, with two distinct rhythmic components: a rapid pulse of ten minutes and slower ultradian oscillations of 50–120 min [22,23]. In type 2 diabetes mellitus (T2DM), impaired pulsatile secretion of insulin has been demonstrated, suggesting a possible mechanism to explain the compromised insulin action in this pathology [23]. A theoretical description of the mechanisms responsible for insulin concentration oscillations has been described, considering the possibility that external stimuli can be necessary to coordinate the behavior of pancreatic islets for generating regular oscillations [24]. 

Insufficient insulin production or insulin activity produces elevated blood glucose levels and causes diabetes mellitus. Type 1 diabetes mellitus (T1DM) is an autoimmune disease that originated from the destruction of β-cells, whereas T2DM is the consequence of a failure of β-cells to produce enough insulin to overcome systemic insulin resistance, frequently associated with obesity, inactivity, and aging. Both types of diabetes also cause cardiovascular issues and alteration in wound repair processes, producing chronic wounds, which can worsen the patient’s clinical condition. Engineered wound dressings have been developed to overcome these difficulties [25,26]. 

## 4. Insulin Receptors and Insulin Binding

Insulin mediates its activities and triggers signaling by binding to the insulin receptor, a transmembrane receptor belonging to the large class of the tyrosine kinase receptors. It consists of a heterotetramer comprising two α and two β glycoprotein subunits connected by disulfide bonds [27]. Insulin binds to the extracellular α subunit, causing the conformational change of the receptor, resulting in the autophosphorylation of a number of tyrosine residues present in the β subunit that increase the kinase activity [28]. This induces tyrosine phosphorylation of intracellular substrate proteins, known as insulin-responsive substrates (IRS) [29]. By binding to other signaling molecules, IRS activates downstream substrates, culminating in the recruitment of glucose transporter 4 (GLUT4) to the cell membrane, which allows glucose uptake [27]. The binding of insulin to its receptor leads, via IRS, to the activation of two different pathways: the phosphatidylinositol-3-kinase (PI3K)-protein kinase B (Akt or PKB) pathway [30] and Ras–Raf mitogen-activated protein kinase (MAPK) pathway, also known as the extracellular signal-regulated kinase (ERK) pathway [29]. The PI3K pathway is responsible for most of the metabolic effects of insulin, while the MAPK pathway is involved in the regulation of gene expression and, by crosstalk with the PI3K and other pathways, cell growth and differentiation control. 

## 5. Insulin Signaling

Besides regulating blood glucose homeostasis, insulin governs gene expression, enzyme activity, nutrient transport, and energy homeostasis. The insulin signaling pathway is the combination of all the proteins involved in the action of insulin and the factors that regulate this pathway. Receptor activation initiates a cascade of phosphorylation and dephosphorylation events, second messenger generation, and protein–protein interaction holding several insulin effects, including activation of transcription factors. In addition, insulin-like growth factor 1 (IGF-1) polypeptide, by binding to the IGF-1 receptor, activates insulin-responsive gene expression, signifying that both insulin and IGF can trigger the same trophic actions [31]. In the brain, IGF-1, indeed, exerts both a metabolic and neuroprotective role by activating the PI3K/Akt and Ras/Raf/MAP pathways. Additionally, it has a strong effect on cellular neuroplasticity, a process associated with learning and memory [31].

The activated insulin receptor induces phosphorylation of IRS proteins in numerous tyrosine residues, some of which are recognized by the Src homology 2 (SH2) domain of the p85 regulatory subunit of the lipid kinase named phosphoinositide 3-kinase (PI3K). At the cell plasma membrane, the p110 subunit of PI3K phosphorylates phosphatidylinositol (4,5) bisphosphate [PtdIns(4,5)P_2_], leading to the formation of Ptd(3,4,5)P_3_. Akt is a key downstream effector of PI3K and Ptd (3,4,5)P_3_, which is recruited from the cytosol to the plasma membrane to permit interaction with other protein kinases that phosphorylate its Thr308 and Ser473 residues [32]. The activation of Akt is a multistep process that requires the protein kinase 3-phosphoinositide-dependent protein kinase-1 (PDK1) to phosphorylate Akt at Thr-308 and another enzyme named PDK2 to phosphorylate Akt at Ser-473 [33]. Activated Akt mediates the regulation of cell growth, proliferation, and the cell cycle by modulating further substrates, such as Gsk3β or FoxOs [34,35]. Further, inhibition of Gsk3β by Akt promotes activation of glycogen synthase, an enzyme that catalyzes the conversion of UDP-glucose into glycogen, contributing to the stimulation of glycogen synthesis [35,36]. 

Another Akt downstream target is the forkhead box O (FoxO) family of transcription factors. These genes show a structural motif, namely the “forkhead box” or “winged helix” domain, which is responsible for binding to chromatin DNA in the nucleus and the activation of target genes [37]. FoxO has an essential role in mediating the effects of insulin on different physiological functions, including cell proliferation, apoptosis, and metabolism. FoxO, depending on its phosphorylation state, plays its role in shuttling between the cytoplasm and nucleus [38]. Under insulin stimulus, activated Akt translocates to the nucleus, where it phosphorylates FoxO at distinct sites and stimulates its interaction with 14-3-3 protein [39]. This chaperone protein promotes the nuclear export and constrains the nuclear import of FoxO protein, resulting in the inhibition of target gene expression that drives the cells to survival [37]. 

PI3K/Akt signaling is responsible for the inhibition of negative regulators of rapamycin mTOR, a Ser/Thr-kinase that is ubiquitously expressed, regulating cell growth and metabolism in response to nutrients, growth factors, and cellular energy conditions. Further, mTOR is composed of two different protein complexes, mTOR complex 1 (mTORC1) and mTOR complex 2 (mTORC2), which differ in some components, responsiveness to rapamycin, as well as upstream and downstream signaling [40]. Their crosstalk is mediated by Akt [40]. Phosphorylated Akt indirectly activates mTOR by inhibiting its negative regulators (tuberous sclerosis complex (TSC1/2)) and activating the mTOR activator, Ras homolog, enriched in the brain (Rheb). Further, mTORC2 phosphorylates Akt at Ser-473 to activate the downstream events and promote Akt degradation. Further, in response to food intake, mTOR directly phosphorylates the insulin receptor, leading to its internalization [41]; this, in turn, induces a decrease in mTOR signaling [42]. 

Further, as already stated, insulin activates the MAPK pathway, which controls a variety of transcription factors and elements, such as cAMP-responsive element-binding protein (CREB), proto-oncogenes c-Myc (MYC), and c-Fos (FOS). Additionally, it helps to regulate the transcription, translation, and post-translational modifications of numerous proteins playing a relevant role in the transduction of extracellular signals to cellular response [43]. 

## 6. Insulin in the Brain

Until 1960, most scientists and clinicians believed that the brain was an organ not sensitive to insulin. At that time, different research groups reported studies simultaneously in which the question of insulin sensitivity was reopened. Rafaelson described the ability of insulin to modulate glucose uptake both in isolated rat spinal cord and in rat cortical brain slices [44,45]. Based on the hypothesis that its action on the peripheral tissues might mask the insulin in the brain, Butterfield and colleagues decided to measure glucose uptake at both sites [46]. For this aim, five healthy male volunteers aged 25–43 were infused with insulin and, after result analyses, it was concluded that the hormone affects both the brain and the peripheral tissues but at a different rate, so the human brain was insulin-sensitive [46]. 

At the beginning, evidence of insulin within the brain was obtained by experiments on dog cerebrospinal fluid (CSF), which denoted the possibility of insulin crossing the blood–brain barrier (BBB) [47]. However, for a long time, it was thought that the presence of insulin in brain tissues and CSF was exclusively derived from pancreatic islets that released it into the blood to be distributed in the target organs. In 1978, Havrankova et al. reported the presence of insulin and insulin receptor (IR) in different rat brain regions, with the highest concentrations discovered in the olfactory bulb and hypothalamus, confirming extrapancreatic synthesis of insulin [48,49]. In addition, studies on rodent models of hyperinsulinemia (obese mice) and hypoinsulinemia (streptozotocin-treated rats) demonstrated that the concentration of insulin in the central nervous system (CNS) was independent of peripheral insulin levels [50]. These data were consistent with the hypothesis that the neurons synthesized insulin in CNS and played a distinct role with respect to peripheral glucose metabolism [50]. Next, further studies showed expression of insulin also in the additional brain area, including the hippocampus, striatum, thalamus, entorhinal, and prefrontal cortices astrocytes [51,52]. More recently, in situ hybridization, ELISA assay, and immunohistochemistry experiments identified the presence of endogenous proinsulin mRNA and protein in immunohistochemistry experiments in pyramidal neuronal cells of the hippocampus and olfactory bulb [53]. Further, insulin mRNA transcripts were detected in human post-mortem brain tissue, and reduced levels were found in the hippocampus and hypothalamus of individuals who had Alzheimer’s disease (AD) [54]. 

Growing evidence suggests that central insulin can have different effects on brain function, so its deficiency can be relevant in some pathological conditions. Insulin can induce endocytosis of AMPA receptors [55] and modulate the transfer of NMDA receptors in the post-synaptic cell membrane, indicating a role in synaptic transmission and plasticity, including long-term potentiation [56]. Notably, many insulin receptors are present both in pre- and post-synaptic compartments, especially in CA1 of the hippocampus region, indicating a significant influence on memory and learning regulation [1]. Furthermore, central insulin regulates glucose homeostasis, maintains energy requirements for different neuronal functions, and is involved in neurite outgrowth and axon guidance through activation of the PI3K/Akt pathway, as demonstrated in different model systems, such as Drosophila [57,58], mice [59], and human neuronal cells [60]. Glucose uptake and utilization in the brain are strictly related to glucose transport. GLUT4 is expressed beside insulin receptors in medial temporal lobe structures associated with cognitive behavior. Insulin activates GLUT4 gene expression and protein shuttling from the cytoplasm to the plasma membrane to regulate glucose uptake and utilization. Further, neuronal glucose uptake depends on glucose transporters 1 (GLUT1) activity in astrocytes and blood–brain barrier (BBB) endothelial cells and by GLUT3 activity in most neurons [3,61]. In addition, insulin and dopamine regulate food intake, reward, and mood in the brain [62]. Nerveless insulin secreted by pancreatic β-cells and transported by cerebrospinal fluid (CSF) can cross the BBB via a saturable receptor-mediated transport, whose efficiency can decrease by pathological conditions, such as obesity, inflammation, and Alzheimer’s disease [2]. Lastly, insulin receptors are expressed in several brain regions but at different levels and densities. A plausible explanation might be that insulin signaling plays a different role in the brain [54]. 

## 7. Brain Insulin Resistance

Insulin regulates food intake and cognitive function in the brain, and defects in insulin signaling may contribute to neurodegenerative disorders. Insulin resistance is typically defined as a failure of body and brain cells to respond to insulin and consists of a reduced ability to stimulate glucose utilization [63,64]. This condition leads to high levels of circulating insulin (hyperinsulinemia) that are associated with hyperglycemia. Cellular insulin resistance is considered an impaired signaling response to insulin. The hormone response can be compromised by downregulation or internalization of insulin receptors or by the failure of insulin binding to the receptors, and, consequently, it can result in insulin signaling impairment [65,66,67]. 

Basically, the brain IR reduces the insulin receptor response, dysregulating the PKT/Akt/FoxO/Gsk3β signaling cascade and compromising cellular homeostasis [67].

Systemic insulin resistance is a syndrome associated with different disorders, such as T2DM, hypertension, and obesity [68]. Growing evidence supports that insulin resistance is the link between T2DM, obesity, non-alcoholic steatohepatitis (NASH) and brain dysfunctions affecting intelligent activity, such as cognition, learning, and memory [69]. In particular, epidemiologic studies demonstrated that T2DM, obesity, and other prediabetic states of insulin resistance are risk factors for Alzheimer’s disease and related disorders [70].

Susan de la Monte proposed a model in which hepatic insulin resistance, caused by metabolic diseases, is a mediator of neurodegeneration, exerting its neurotoxic effect via a liver–brain axis [69]. Toxic lipids, including ceramides, are produced in the liver and released into circulation due to their lipid-soluble nature, cross the BBB, and produce a neurodegenerative effect by impairing insulin signaling. Ceramides are lipid signaling molecules that increase in the liver with hepatic insulin resistance, probably due to the elevated biosynthesis or reduced degradation [71]. Further, cytotoxic ceramides in the brain induce oxidative stress, ER stress, mitochondrial dysfunction, lipid peroxidation, new ceramides synthesis, and inflammation, mechanisms leading to neuronal death [69]. Nevertheless, based on the association between metabolic disorders and brain insulin resistance and deficiency, it has been coined type 3 diabetes (T3D) [72]. T3D is a specific brain condition in which insulin resistance induces cognitive impairment and neurodegeneration. Similarly, it was suggested that the term “diabesity”, to describe the simultaneous presence of T2D and obesity and how these morbidities can contribute to age-related cognitive dysfunction, including AD, has been explored [73]. The concept of T3D was demonstrated by a study using rodents in which intracerebral administration of streptozotocin (STZ) induced an AD-like cognitive impairment [74]. These findings were validated by clinical studies on AD patients in which significant reductions in insulin and IGF-1 receptor levels in the frontal cortex, hippocampus, and hypothalamus were found [75]. 

In our laboratory, a tight relationship between metabolic and neurodegenerative disease was revealed by using a mouse animal model of diet-induced obesity and insulin resistance. In the brain of obese mice, a reduced level of the insulin receptor and impairment in insulin signaling associated with the presence of apoptosis and amyloid plaques formation were found [6]. Furthermore, the use of specific biomarkers showed an increase in oxidative stress and inflammation. These insulin-resistance-induced alterations were associated with the disorder in fission and fusion, the mechanism involved in mitochondrial dynamics and a feature of mitochondrial dysfunction [6].

Further, evidence of a correlation between dysmetabolism and AD was obtained in obese insulin-resistant rats and rodent models of AD. Brain insulin resistance induced activation of cytokines, such as IL-1, IL-6, and TNFα, chemokines, and pro-inflammatory enzymes, such as COX-2 and iNOS, in the hypothalamus and cerebral cortex in other brain structures, such as the cerebellum, amygdala, and hippocampus [76]. 

On this indication, it could be extrapolated that therapeutic strategies designed to treat T2DM, obesity, or dyslipidemic disorders, and systemic insulin resistance could help to prevent or reduce neurodegeneration. In agreement, several studies have reported that treatment with a hypoglycemic or insulin sensitizer agent can exert a protective role, reducing the incidence or severity of AD [77,78]. However, the association between antidiabetic medication and the risk of neurodegenerative diseases is conflicting [79,80,81,82,83,84,85]. In particular, outcomes regarding the biguanide metformin indicate that its neuroprotective role can be performed only in combination with insulin [81,82]. Moreover, some reports suggest that GLUT4 plays a relevant role in hippocampal memory processes and dysregulation of this transporter could cause cognitive impairments related to insulin resistance [86]. Hence, dysregulation of GLUT4 could, at least in part, explain the comorbidity of T2D and AD diseases. Furthermore, several findings indicate that insulin resistance in the brain of obese mice can be inhibited or reversed by feeding animals with a diet in which regular intake of pistachio or honey or natural dietary supplements are consumed [87,88,89,90,91]. Diet, indeed, has become the object of investigation concerning cognitive aging and neurodegenerative disease. High consumption of single nutrients, functional food, or a Mediterranean diet rich in antioxidant molecules associated with exercise could be a strategy for preventing insulin-resistance-induced neurodegeneration [92]. Considering the role of insulin signaling in the brain, it is possible to establish that a pathological condition induced by insulin resistance can be associated with cognitive and metabolic alterations.

## 8. Alzheimer Disease and Insulin

Alzheimer’s disease (AD) is the most diffused cause of dementia (60/70%) that is progressively increasing in the aging world population and severely impacts the quality of life of both patients and caregivers. Diagnosed for the first time by the German neuropathologist Alois Alzheimer in 1906, it is asserted as the sixth major cause of death globally. Patients with AD gradually decline in cognitive abilities and memory functions [93]. Currently, the available therapies for AD cannot decelerate or prevent the progression of the disease. AD clinical classification comprises two subtypes: late-onset or “sporadic” AD (SAD), and early-onset or “familial” AD (fAD). SAD includes 95% of patients aged 65 years or older, while 5% of patients develop FAD before 65 years of age since they carry rare genetic mutations in presenilin 1 (PS 1) and presenilin 2 (PS 2) or amyloid precursor protein (APP) genes [94].

The histopathological hallmarks of AD are extracellular senile plaques of insoluble amyloid-beta (Aβ) peptide and intracellular neurofibrillary tangles (NTF) of hyperphosphorylated tau protein, which causes microtubule disassembly [95].

Beta-amyloid is a peptide of 40–42 amino acids derived by the abnormal proteolytical cleavage of APP. Accumulation of Aβ leads to native conformation change, and it acquires a beta-sheet motive-promoting protein aggregation that forms fibrils and amyloid plaques [96,97]. Inhibition of fibrillogenesis represents a reasonable target for AD prevention and therapy. However, some findings indicate that smaller Aβ-soluble oligomers can be more toxic than amyloid plaques [98]. Based on Aβ toxicity, for several years, the amyloid cascade hypothesis has been assumed as the main cause of AD onset [99]. According to this hypothesis, the accumulation of Aβ induces a biochemical, histological, and clinical change in AD patients. Recently, a monoclonal antibody has been developed just against beta-amyloid (aducanumab), capable of reducing beta-amyloid plaques and, reasonably, clinical AD decline. In June 2021, aducanumab was approved for the treatment of Alzheimer’s disease by the US Food and Drug Administration (FDA) [100].

Further, the APP gene is highly conserved during evolution, and the use of vertebrate or invertebrate simple model systems has facilitated the understanding of several biochemical mechanisms underlying AD [101,102,103]. 

At the cellular level, AD is characterized by a progressive loss of pyramidal cells in the entorhinal cortex and CA1 region of the hippocampus, the area responsible for maintenance of higher cognitive functions, such as learning, orientation, and language disturbance, and recognizing or identifying persons. [104]. AD’s early symptoms are also marked by synaptic dysfunction that disturbs neuron–neuron communication, thereby initiating the gradual loss of memory. Accumulating evidence suggests that mitochondrial dysfunction, an early pathological event, is a key factor in synaptic damage and correlates with cognitive deficits and memory loss [105]. Therefore, based on these results, the mitochondrial cascade hypothesis has been formulated, which is regarded as one of the earliest features in AD [106]. Further, Aβ oligomers can bind or enter cellular organelles or compartments, mediating dysfunctions, leading to synaptic loss and cell death [107].

Although aging is the main risk factor for AD, it has been observed that patients with T2D, obesity, or other metabolic diseases develop reduced cognitive performance and have the greatest risk of developing AD [108]. Several findings indicate that insulin plays a neuroprotective and trophic function, thus countering beta-amyloid toxicity, oxidative stress, and inducing neuronal survival. Further, brain insulin resistance associated with impaired cerebral glucose metabolism has been reported in Alzheimer’s disease [67,109]. As a consequence of metabolic disease, IR leads to loss of synapses, impaired autophagy, and increased neuronal apoptosis. These modifications could trigger a cascade of events leading to abnormal Aβ and tau accumulation, culminating in Alzheimer’s disease pathology [110]. This was in agreement with a study showing that peripherical insulin resistance was correlated, in AD patients, with brain Aβ deposition in the frontal and temporal areas [111]. 

The link between Aβ toxicity and insulin was elucidated by in vitro studies. Using hippocampal neurons demonstrated that administration of Aβ-derived diffusible ligands (ADDLs) induced the loss or internalization of insulin receptors, causing oxidative stress and synaptic spines deterioration [66]. In addition, in neuroblastoma cells, Aβ oligomers redistributed or reduced the insulin receptor numbers on the cell membrane, inhibited Akt activation, increased oxidative stress, and induced mitochondria dysfunction and apoptosis. Insulin administration reverted all these dysfunctions, indicating a neuroprotective role [67]. Further, apoptosis was prompted by inhibiting shuttling of phosphorylated Akt from the cytoplasm to mitochondrion, where cytochrome c release and caspase activation were inhibited. However, in both these studies, insulin addition prevents all the Aβ-induced dysfunctions, and insulin signaling provides a physiological defense against Aβ toxicity. An explanation of these results might be attributed to a competition between insulin and Aβ for the binding to insulin receptors and for the interaction with the protease insulin-degrading enzyme (IDE) [65,112]. Aβ is able to bind to insulin receptors, causing deficiency in the insulin signaling cascade and inhibition of the cell survival pathways. Hyperinsulinemia leads to the antagonism between insulin and Aβ for IDE, modulating the Aβ brain clearance.

## 9. Intranasal Insulin as an AD Therapeutic Agent 

Brain insulin has a neuroprotective function against the accumulation of senile plaques by regulating Aβ peptide levels and preventing the binding of Aβ to the synapses [7,65,66]. Thus, restoring brain insulin function in brain insulin resistance conditions may provide therapeutic benefits for adults with AD. Based on this knowledge, the idea of using insulin as a potential therapeutic agent in neurodegenerative diseases has been developed. Nerveless, peripheral administration of insulin is not a viable route due to the risk of systemic hypoglycemia or induction/increase in insulin resistance also in individuals who do not have diabetes. To overcome this problem, an intranasal approach has been proposed as a potential strategy to deliver the hormone directly to the CNS [113,114]. Due to the high density of insulin receptors and the direct connection between the nasal cavity and the brain structures, insulin can be delivered along olfactory and trigeminal perivascular channels by bypassing the BBB. The possibility of being used for AD treatment has been demonstrated by a study in rats or by proof-of-concept, as well as pilot clinical investigations [113,115]. A follow-up study (SNIFF) in which 20 IU or 40 IU of insulin or placebo were intranasally administered to 104 subjects with AD or amnestic mild cognitive impairment (MCI) for 4 months was conducted by Susan Craft and colleagues [116]. As indicated by the memory test, both doses improved memory performance, and FDG-PET measures showed a minimal increase in plasma insulin or glucose level [116,117]. 

In light of these encouraging findings, a randomized study with AD and MCI participants was conducted to determine insulin’s feasibility, safety, and efficacy [118]. Indeed, 40Iu of intranasal insulin or placebo were administered to AD and MCI individuals per day for 12 months, followed by a 6-month open-label extension. Using the ADAS-Cog12, a test measuring cognitive function, mood, and behavior, the change in memory performance from the baseline to 12 or 18 months was detected. Then, the change in CSF biomarkers (e.g., amyloid-beta 42 [Aβ_42_] and amyloid-beta 40 [Aβ_40_], total tau protein, tau p-181, insulin concentration) was examined. In order to improve intranasal insulin delivery, a device (device 1) was successfully used in previous trials, and an alternative device (device 2) was utilized. After one year of treatment, no cognitive or functional difference between the insulin and placebo groups in the 240 cohort who used device 2 was found [118]. Conversely, at 12 months, the device 1 cohort (49 participants) showed improvements in memory performance for the insulin group compared with the placebo group. During the six-month open-lab period, the participants using device 1 continued to do better than those in the placebo group [118]. 

The authors discuss that the discrepancy could be due to the device 2 malfunction, not previously tested in AD trials, and that the access of insulin to the CNS needs additional measures. Further, a well-tested delivery device should be designed for the intranasal delivery of therapeutic agents [117,118]. Sequentially, a randomized study with a group of 78 participants demonstrated, after one year, that intranasal insulin reduced the accumulation of white-matter hyperintensity volume in association with improvements in cognition in the insulin-treated group compared to placebo patients [117].

Although positive effects were detected in intranasal insulin, some side effects, such as irritation and damage of the nasal mucosa [119] or an increase in arterial blood pressure [120], were revealed. Furthermore, studies evidenced that insulin could not penetrate effectively into the brain after intranasal administration without designing a delivery strategy [121]. For this aim, a nanogel as a smart insulin delivery system was engineered and named NG-In [122]. The biocompatibility and hemocompatibility properties of the nanogel were tested, together with the ability to protect insulin by protease degradation. NG-In was able to bind to insulin receptors and activate the Akt pathway. Additionally, by in vivo study fluorescent, NG-In was administered through the intranasal route in mice and the biodistribution, clearance, delivery to different brain areas, and its biological activity were measured [123]. The outcomes indicated that the use of NG-In might be a promising new approach to restore cerebral insulin function in insulin resistance conditions.

Likewise, patients with peripheral insulin resistance due to obesity or T2D showed a reduced response to intranasal insulin in the CNS, as measured by functional neuroimaging methods [124,125]. Moreover, improving peripheral metabolism in obese and diabetes subjects through dietary restrictions has been shown to restore CNS insulin sensitivity, and treating CNS insulin sensitivity improves peripheral metabolism [125]. However, although the field of nose-to-brain insulin administration still remains to be explored, it continues to be the main promising approach for insulin brain delivery. 

## 10. Insulin, Insulin Resistance, and Mitochondrial Dysfunction

Insulin regulates cellular metabolism and controls nutrient homeostasis through the IRS/PI3K/Akt signaling cascade and FoxO and mTOR effectors. MTORC1 signaling is critical for regulating protein, lipid, and fatty acid synthesis, as well as mitochondrial metabolism. FoxO, instead, promotes cell survival and proliferation. Since mitochondria are an important metabolic platform, they undergo dysfunction under insulin resistance in metabolic and related neurodegenerative diseases. Several data, published by different research groups and ourselves, have provided evidence that insulin resistance and its cerebral effects are strongly associated with the alteration of mitochondrial homeostasis [11,126]. However, the molecular link between insulin resistance and mitochondrial dysfunction has not yet been well defined. In the following sections, we will review recent studies linking insulin action with mitochondrial metabolism and function. 

Mitochondria are interconnected organelles that play a critical role in various cellular processes, including: ATP production by oxidative phosphorylation (OXPHOS), metabolite biosynthesis, intracellular calcium buffering and signaling, ROS generation, steroidogenesis, immune responses, β-oxidation of fatty acids, modification of phospholipids, apoptosis, and stem cell reprogramming [127].

The structure of mitochondria consists of an outer mitochondrial membrane (OMM), which separates the intermembrane space from the cytosol and is generally permeable to ions and larger molecules, and a much less permeable inner mitochondrial membrane (IMM), forming cristae into the matrix [128]. Across the IMM, an electrochemical gradient is formed, which allows ATP production through oxidative phosphorylation, heat formation, proton escape, and ROS generation through oxygen reduction [128]. Thus, mitochondria are the central hub to convert energy for cellular processes. In addition, mitochondria can interact with other organelles, such as the endoplasmic reticulum (ER), the lysosomes, and exchange materials, operating as a signaling platform [128]. 

Mitochondria are highly dynamic organelles with the ability to change size, shape, position, and to divide and fuse on the basis of two well-balanced processes called fission and fusion. Beyond these changes, mitochondrial mobility through the cytoskeleton is critical for mitochondria’s cellular distribution and turnover [128,129].

In the brain, a tissue with high energy demand, neuronal cells, are highly dependent on mitochondrial presence and function. Although it represents only 2% of the body’s mass, it consumes approximately 20% of the entire body’s energy. The deficit in mitochondrial ATP production reduces energy metabolism and influences glucose intake in neurons. Failure of glucose metabolism in the brain is a cause of aging and neurodegenerative disorders [130]. 

However, in neurons, mitochondrial function is not limited to energy provision but also to establishing membrane excitability and performing the complex processes of neurotransmission and plasticity. It is clear that maintaining a healthy mitochondrial population is critical for neuronal health. Consequently, it is essential to understand the molecular mechanisms underlying mitochondrial dysregulation and its potential implications for neurodegeneration.

Different neurodegenerative diseases, including Alzheimer’s disease (AD), Parkinson’s disease (PD), Huntington’s disease (HD), and amyotrophic lateral sclerosis (ALS), are characterized by the progressive loss of selective neurons in the CNS. More studies report mitochondrial dysfunction as a common feature of these neurodegenerative diseases, manifested by ATP deficiency, overproduction of ROS, mitochondrial-driven inflammation, and an increase in proapoptotic molecules, all dysfunctions leading to programmed cell death [131].

Glucose is the primary source of energy used by mitochondria to make ATP. It is transported to neurons via the 3 transporter (GLUT3) and, after conversion to pyruvate, enters into mitochondria to produce ATP via the Krebs cycle. An essential role in this process is played by Akt, a key molecule in insulin signaling. Akt, on the one hand, regulates the expression of GLUT3 and its translocation to the plasma membrane; on the other hand, it translocates to the mitochondria to increase the binding of the glycolytic enzyme hexokinase II (HKII) to the mitochondrial surface and promote glycolysis [132].

Mitochondria are truly interesting organelles for their adaptability to metabolic condition changes, having developed a mitochondrial quality control (MQC) system to overcome the condition of damage that often underlies neurodegenerative and metabolic diseases. 

Several mechanisms contribute to MQC: (a) a balance of fusion and fission, processes necessary to maintain morphology and volume of mitochondria; (b) biogenesis that ensures a sufficient number of functional mitochondria; (c) mitophagy, the mechanism necessary to remove the damaged organelles, whose discussion will be better addressed in the following sections [131].

Neurons, as stated before, are largely dependent on mitochondrial activity and energy production to permit synaptic transmission and reuptake of neurotransmitters. When the cerebral mitochondria cannot satisfy the energy demand in neuronal cells, a series of degenerative events is triggered whose consequence is loss of cognitive and memory functions [133]. The intensity of mitochondrial functional deterioration and energy dysmetabolism is in some way correlated with the course of the disease. Therefore, the maintenance of the mitochondrial bioenergetic functions could prevent these alterations. 

Several mechanisms contribute to mitochondrial dysfunction, including mutation in mtDNA, decreased mitochondrial biogenesis, increased fission, decreased OXPHOS activity, dysfunction in the mitophagic process, altered bioenergetics, and imbalance of calcium homeostasis [131]. One of the consequences is the decreased oxidation of substrates, including fatty acids, which causes lipids accumulation. An increased level of lipids induces deposition of their mediators as diacylglycerols (DAG) and ceramides (CER) that, in turn, inhibit insulin signaling [134]. DAG, through the activation of the protein kinase C, translocates to the plasma membrane and inhibits the insulin receptor, whereas CER interferes with the insulin signaling at the level of protein kinase Akt [134].

Another important consequence of mitochondrial dysfunction is the increase in reactive oxygen species (ROS), which damages cellular and mitochondrial components and compromises mitophagy by amplifying cell damage up to apoptosis [134]. Some studies carried out on adipocytes in myotubes show that high levels of ROS cause a reduction in the insulin metabolic effect [135].

Thus, functional neuronal insulin/IGF1-1 signaling is essential for healthy metabolism in the brain and is closely related to mitochondrial energy homeostasis [136]. In agreement with these findings, activation of the insulin/IGF-1 pathway improves mitochondrial respiration, ATP production, AMPK activation, Ca^2+^ buffering, and increased PGC1α expression, revealing the importance of the interplay between mitochondrial function and insulin/IGF-1 action to maintain brain activity [136].

Moreover, beyond neurons, a close relationship between insulin signaling, mitochondrial dysfunction, and activation of microglia exists. Insulin can act on astrocytes and glial cells, influencing their metabolism. In glial cells, in fact, insulin, by binding to its receptor, activates the IRS-1/PI3K/Akt signaling that stimulates phagocytic action and proinflammatory activity of microglia and influences the mitochondrial production of ATP [136]. In mice, a high-calorie diet induces, in proopiomelanocortin (POMC) neurons, activation of microglia, which produces TNFα and stimulates mitochondrial ATP production. This activation promotes mitochondrial fusion and contributes to obesity development [137].

In addition, astrocytes are insulin-responsive cells, and, here, insulin signaling controls the response of mitochondria to the availability of nutrients and regulates the systemic insulin and glucose tolerance [138]. Further, knockout of the insulin receptor in astrocytes causes a reduced release of glial ATP, with a consequent decrease in dopamine release. This mechanism could be responsible for anxious and depressive behaviors in people with insulin resistance disorders, such as diabetes and obesity [139].

Furthermore, insulin exerts an antiapoptotic effect in the CNS due to its antioxidant properties and the ability to reduce the release of mitochondrial cytochrome c via Akt activation. This molecule translocates and accumulates in the mitochondria phosphorylates, together with the beta subunit of ATP synthase, the glycogen synthase kinase-3beta (Gsk3β), and the Hexokinase II (HKII) [140,141]. Insulin, indeed, regulates mitochondrial metabolism and oxidative capacity through PI3K/Akt signaling [140,141]. The decreased Akt signaling that occurs in IR conditions leads to a reduction in mitochondrial respiration and increases in mitochondrial fission, causing significant alterations in mitochondrial efficiency and function and a consequent increase in oxidative stress [140,141].

Several investigations confirm the active role played by insulin in the mitochondria. Ruegsegger et al., for example, report decreased ATP production, citrate synthase, and cytochrome oxidase activity in the cerebellum, hypothalamus, and hippocampus brain regions of streptozotocin-induced diabetic mice [142]. In contrast, intranasal insulin administration increases mitochondrial ATP production and improves neuronal development and neurotransmission. Thus, this outcome indicates a direct regulatory role of insulin on mitochondrial function and suggests its potential therapeutic benefits in brain insulin resistance conditions [143]. In addition, insulin and IGF-1 treatment on striatal cells, derived from Huntingtin knock-in mice, with impaired mitochondrial function, decreased mitochondrial ROS generation and mitochondrial Drp1 phosphorylation. This indicated that the fission mechanism was reduced and that mitochondrial function was ameliorated in a PI3K/Akt-dependent manner [144].

The idea of a central role of insulin in mitochondrial homeostasis was also reinforced by our team [145]. Using a cellular IR model, we observed that treatment with insulin enabled the recovery of Aβ-induced mitochondrial dysfunction [145]. Furthermore, several reports indicate that chronic high-fat feeding (HFD) causes insulin resistance in brain mice. Indeed, in the hypothalamus of HFD mice, the association of insulin resistance to dysregulated mitochondrial homeostasis was demonstrated by altered morphology, decreased mitochondrial respiration, and reduced mitochondrial-endoplasmic-reticulum contact sites, and disturbances in the quality and number of mitochondria [136]. Moreover, our studies using the same animal model showed an alteration in the mitophagic process in a time-dependent manner associated with the condition of insulin resistance [145].

In summary, these data confirm the significant role of insulin and its signaling on mitochondrial function and the importance of this interaction for metabolism and brain functions. It is conceivable that an improvement in insulin signaling could overcome dysfunction in the brain in both metabolic and neurodegenerative diseases. 

Let us now detail the individual mechanisms that contribute to mitochondrial quality control (i.e., dynamics, mitophagy, and biogenesis) fundamental for cellular health. 

## 11. Mitochondrial Dynamics

Mitochondria form elongated tubules that continually divide (fission) and fuse (fusion) to form a complex, well-organized, and highly dynamic network inside cells. The number, morphology, size, and placement of mitochondria change in response to pathophysiological signals and are regulated by fusion and fission events. The balance between these processes ensures proper mitochondrial functioning by adapting the network to the availability of nutrients and the cell’s metabolic state, which is referred to as “mitochondrial dynamics” [128].

Fusion occurs when two adjacent mitochondria join together, forming a hyperfused network with elongated and highly connected mitochondria to mix their contents, including mitochondrial DNA (mtDNA) and metabolic intermediates, to recover the activity of their damaged or depolarized membranes. On the other hand, fission leads to mitochondria’s fragmentation, increases their number, and prepares the cell for its division and meiosis, and, as discussed later, facilitates the autophagic clearance of dysfunctional mitochondria by mitophagy [128].

The molecular mechanisms of fission and fusion have been widely examined in detail [146], and, here, we will briefly describe the mechanisms that control these processes. Mitochondrial fusion and fission are regulated by proteins present or translocated in the inner or outer membranes of the mitochondria or that are soluble in the inter-membrane space. The main proteins involved in fusion/fission processes are guanosine triphosphatases (GTPases), a large family of hydrolase enzymes that bind to the nucleotide guanosine triphosphate (GTP) and hydrolyze it into guanosine diphosphate (GDP) [146].

In mammals, fusion is driven by a two-step process mediated by dynamine-like GTPases, which, upon hydrolysis, undergo a conformational change that carries adjacent mitochondrial membranes closer together, creating contacts between them [147]. Mitofusin1/2 (MFN1/2) coordinates the fusion of the OMM, while optic atrophy 1 (OPA1), cooperating with cardiolipin, is responsible for the fusion of the IMM [148]. Furthermore, MFN2 is also a key regulator of endoplasmic reticulum (ER)–mitochondria tethering, which are contact sites highly involved in insulin signaling and, as discussed below, implicated in the mitochondrial constriction of the fission process [128,149,150].

Mitochondrial fission is a multi-step process leading to the division of a single mitochondrion into two daughter mitochondria. The GTPase dynamin-related protein drives mitochondrial fission (Drp1), a cytosolic protein, which, upon recruitment to the OMM, oligomerizes and forms a ring structure around the mitochondria, leading to membrane shrinkage and mitochondrial division in a GTP-dependent manner (Figure 1) [128,150,151].

Drp1 has a pleckstrin-homology (PH) domain that mediates binding at membrane phospholipids. Its recruitment to the OMM needs the scission of multiple membrane-anchored adaptors, including Fis1, tail-anchored proteins, mitochondrial fission factor (MFF), and mitochondrial dynamics proteins 49 and 51 (MiD49 and MiD51) [152].

Interestingly, although Fis1 is not involved in the fission mechanism under basal conditions, its overexpression can induce mitochondrial fragmentation in the absence of DRP1 [153] by activating fission and blocking the fusion machinery by preventing the GTPase activity of MFN1, MFN2, and OPA1 [154].

Mitochondrial dynamics also include the involvement of the plasma membrane and organelles, such as ER and lysosomes. The ER–mitochondria contact points, referred to as mitochondria-associated ER membranes (MAM), are the sites where the exchange of Ca^2+^, lipids, and metabolites takes place. An increasing number of papers suggest that the integrity of MAM is required for insulin signaling [155,156]. ER is also required for the initial step of the mitochondrial division [146]. ER tubules, indeed, not only make contact with mitochondria but also wrap around them, leading to mitochondrial constriction [146]. The discovery that MAMs constitute the site of fission and that, before mitochondrial division, Drp1 and its adapters, MFF and MiD49/51, are specifically recruited to these ER–mitochondria contact sites opened the way to new hypotheses on the mechanism of the IMM division [146]. Recent works have suggested that IMM constriction is Ca^2+^-dependent and occurs at ER–mitochondria contact sites [107,108] and can induce IMM division before Drp1 recruitment [128].

Recently, an unexpected organelle has been identified as a new player in regulating mitochondrial fission. The formation of a mitochondria–lysosome membrane has been identified, in HeLa cells, by electron microscopy and confocal live-cell imaging measures, suggesting that lysosomes can have a role in mitochondrial dynamics via RAB7 GTPase [157]. Organelle dynamic and morphological transitions control cell fate decisions. Thus, it represents a crucial step for understanding the mechanism underlying human diseases, especially those associated with mitochondrial, ER, and lysosomal dysfunctions. The effect of insulin/insulin resistance in organelle dynamics will be addressed in the next section. 

## 12. Mitochondrial Dynamics and Insulin

The highly dynamic behavior of mitochondria allows them to change their morphology and intracellular distribution on the bases of the physiological conditions and energy demands of a given tissue at a given moment and requires a balance between fission and fusion. Loss of this balance provokes a series of events, such as excessive ROS release, mitochondrial dysfunction, and altered metabolism, which contribute to the pathogenesis of metabolic diseases and related neurodegeneration, for example, in mice, the knockdown of fusion proteins, altered homeostasis promotes insulin resistance, and obesity [158]. On the other hand, genetic ablation of MFN1, fusion protein, in the liver led to enhanced lipid use, increased mitochondrial respiratory capacity, and the formation of a highly fragmented mitochondrial network [159]. Further, mice lacking the mitochondrial fission protein Drp1 showed, in the liver, protection from high-fat-diet-induced insulin resistance and obesity [160].

Additional data support the involvement of mitochondrial dynamics processes in the pathogenesis of metabolic disorders. As in obesity or type 2 diabetes, exposure to an excess of nutrients promotes mitochondrial fragmentation and decreases mitochondrial fusion, which, instead, increases under hunger or stress conditions. [161]. The hippocampus of the diabetic mice model shows changes in the number and morphology of mitochondria, together with decreased functional capacity, evidenced by lower production of ATP [161]. Furthermore, in the rat liver cell line or myoblast, after exposure to high concentrations of glucose, mitochondria increase the fission process and undergo fragmentation associated with ROS overproduction [162].

Several decades of intense scientific research have revealed that Drp1 is enriched at nerve endings and is involved in synapses formation. Drp1 hyperactivation, related to diabetic insult and neurodegenerative diseases, causes a wide range of damage, including mitochondrial depolarization, oxidative stress, bioenergetic defects, and increased mitochondrial fission in the brain, leading ultimately to cell death [163]. Inhibition of Drp1 activation is protective in several neurodegenerative diseases, such as Parkinson’s disease, Huntington’s disease, and multiple sclerosis [164].

Increased levels of Drp1 and an imbalance in mitochondrial dynamics have been found in diabetes and obesity conditions [165]. Alteration of Drp1 is associated with synaptic damage, and its inhibition protects against synaptic dysfunction in the hippocampus of diabetic or AD animal models, significantly improving learning and memory [166]. Furthermore, the modulation of Drp1, observed in the hippocampus of a mouse model of diabetes by Gsk3β, confirms the role of insulin signaling in mitochondrial dynamics. Modulation of the Drp1 pathway or inhibition of Gsk3β activity restores, indeed, the impaired long-term potentiation in the hippocampus of db/db mice [166]. 

Recently, Park et al. assembled pieces of evidence about the role played by mitochondrial fission proteins in diabetic neurons [163]. They report that the hippocampus of diabetic mice, with a deficiency in neuron-specific Drp1 (Drp1cKO), showed changes in mitochondrial morphology and distribution, oxidative stress, dendritic spines, and synaptic loss damage. Therefore, modulation of proteins involved in neuronal mitochondrial dynamics could provide a therapeutic target against synaptic damage. Further, in diabetic neuropathy, the presence of small and fragmented mitochondria associated with increased expression of Drp1 was discovered. A reduction in aberrant mitochondrial fission by Drp1 knockdown resulted in reduced susceptibility to hyperglycemic damage and increased neuronal survival [167]. Although further investigations are needed, the existing evidence confirms that dysfunction in mitochondrial dynamics plays an essential role in insulin signaling and regulation of glucose metabolism. This process can be considered an early starting point for metabolic diseases and connected neurodegeneration. Thus, modulating the levels of fission and fusion mediators may represent an interesting therapeutic approach. 

## 13. Mitophagy 

Maintaining a healthy mitochondrial population is critical for cell viability. As described before, mitochondrial dynamics ensure that the correct morphology and size of the mitochondria and their quantity and quality are regulated by biogenesis and mitophagy processes. 

Biogenesis and mitophagy are, in fact, intimately connected by proteins such as SIRT1, which activates PGC-1α and stimulates autophagy mediated by various deacetylating autophagic proteins (e.g., Atg7, Atg5, and LC3) [168]. Another regulator of autophagy is mTOR, which controls mitochondrial biogenesis through the activation of PGC-1α and its target genes. There are several mechanisms built by the cells to maintain mitochondrial homeostasis, including the proteolytic system and the proteasome. In addition, under stress conditions, the formation of the vesicle derived from a mitochondrion can be degraded by a lysosome, a topic not addressed in this review. Here, we will focus on the most studied system for removing damaged mitochondria, based on a very well-regulated process of selective autophagy called mitophagy [168]. This process prevents the accumulation of abnormal or damaged mitochondria and promotes the maintenance of a stable number of healthy mitochondria within cells. Autophagy is a widely studied and evolutionarily conserved catabolic pathway used by the cell to eliminate proteins or organelles that are no longer necessary. Thus, damaged organelle clearing guarantees cell and tissue integrity. This cellular degradation process requires more than 40 autophagy-related genes (denoted ‘Atg’) proteins [168]. During autophagy, a double membrane vesicle, called the autophagosome, engulfs harmful or misfolded or aggregated proteins or damaged organelles, leading them to fuse with the lysosome. Here, proteolytic enzymes facilitate protein/organelle degradation, whose products are released back into the cytoplasm, where they are recycled to build new macromolecules [169]. Thus, mitophagy selectively degrades damaged mitochondria, contributing to MQC (Figure 1).

In mammalian cells, two main mechanisms of mitophagy signaling have been identified: (a) the PINK1/Parkin-independent form; (b) the receptor-mediated mitophagy, based on receptors located in the OMM. Among the receptor mitophagy, we can distinguish the BNIP3/NIX-mediated mitophagy also involving BCL2L13, prohibition 2 (PBH2), and the FUN14 domain-containing 1 (FUNDC1)-mediated mitophagy [170,171,172]. All these receptors contain an LC3-interacting region (LIR) domain. The best-characterized ubiquitin-mediated mitophagy depends on Ser/Thr-kinase PTEN-induced putative kinase 1 (PINK1) and E3 ubiquitin ligase Parkin, which activates mitophagy through indirect binding to autophagosomes [173]. In contrast, BCL-2/adenovirus E1B interacting protein 3 (BNIP3) and Nip-like protein X (NIX) activate mitophagy under hypoxia or nutrient deprivation through direct binding to autophagosomes and play an important role in mitophagy-mediated mitochondrial quality control. Further, under hypoxic conditions, FUNDC1 directly binds to LC3 through its LIR motif, inducing mitophagy [174].

Mammalian mitophagy depends on PINK/Parkin signaling [175] (Figure 1). In healthy and polarized mitochondria, PINK1 is constitutively transported by the translocase complex of the TIM/TOM mitochondrial membrane within the organelle, where it is cleaved initially by the mitochondrial processing peptidase (MPP) and successively by the membrane protease presenilin-associated rhomboid-like protein (PARL) [176]. Mitochondria then dissociate the PINK1 double-cleaved form (exposing F104 at the N-terminus) and rapidly degrade within the cytosol. In aged or damaged mitochondria, the loss of mitochondrial membrane potential (ΔΨm) disrupts the standard proteolytic processing of PINK1. It prevents its import into the inner membrane, leading to the accumulation of PINK1 on the outer mitochondrial membrane (OMM). Here, to become a highly active kinase capable of promoting Parkin translocation to mitochondria through phosphorylation at Ser-65 and its ubiquitination, pSer65-Parkin is recruited into mitochondria, which changes conformation and activates its ubiquitin ligase activity [177]. Specific proteins located on the OMM, such as the voltage-gated anion channel (VDAC), the family members of the homolog Ras T1 (RHOT1), and MFN1/2, act as substrates of Parkin’s phospho-ubiquitin [178]. Polyubiquitinated mitochondrial proteins recruit several autophagy adapter molecules, such as the neighbor of BRCA1 (NBR1), sequestosome-1 (p62/SQSTM1), and optineurin (OPTN), TAX1 binding protein 1 (TAX1BP1), and NDP52 (nuclear dot protein 52kDa). These adapter proteins contain a ubiquitin-binding domain (UBA) and a microtubule-associated protein 1 light chain 3 (LC3) interacting region (LIR), which serves as an anchor for developing an autophagosomal membrane around the tagged mitochondrion [179]. Fusion with lysosomes completes the organelle degradation [180]. 

As discussed, PINK/Parkin-pathway-mediated mitophagy is the basis of intracellular homeostasis and is a key point in mitochondrial quality control [181]. Therefore, it is easy to guess that defects in this process are widely associated with many diseases, including neurodegenerative diseases associated with brain insulin resistance, such as AD [133].

Several studies have shown a link between mitophagy and mitochondrial dynamics. The increase in mitochondrial fission is considered one of the first steps in mitophagy induction (Figure 1). 

Recent evidence suggests an additional role of Fis1, a protagonist in mitochondrial fission, in mitophagy [180]. Upregulation of both fission and mitophagy processes, mediated by Fis1, have a pathological role in diabetes mellitus [181]. Similarly, the overexpression of Drp1 and the consequent activation of mitochondrial fission facilitate mitochondria’s fragmentation and elimination when a high-fat diet is followed [180]. In INS1 cells, inhibition of fission, induced by Drp1 or Fis1 silencing, reduced mitophagy and resulted in the accumulation of oxidized mitochondrial proteins and decreased insulin secretion [182]. 

Finally, it is also interesting to mention MFN2, which, besides regulating mitochondrial fusion and participating in the formation of MAM between the ER and mitochondria, is also able to interact with PINK1 and Parkin1 [183]. Indeed, in mammals, it has been observed that, when PINK1 phosphorylates MFN2 at Ser-378, it modifies its conformation and blocks the fusion process [184]. In contrast, when MFN2 is phosphorylated at Ser-442 and Thr-111, it acquires the ability to bind Parkin and activate mitophagy and the clearance of mitochondria [185]. Failure of mitophagy for abnormality of the lysosomal system leads to the accumulation of damaged mitochondria that are not removed. This impairment activates programmed cell death, which is the basis of neurodegenerative disease.

## 14. Mitophagy and Insulin

Mitochondrial dysfunction is implicated in numerous pathologies, both of metabolic and neurodegenerative origin; thus, it is not surprising that there is a tendency to assign a leading role to mitophagy. Numerous data correlate the dysfunction of this clearance process with the development of insulin resistance conditions, present both in metabolic syndromes, such as T2DM, obesity, and hyperlipidemia, and neurodegenerative diseases, such as AD [181,182,183,184].

Mitophagy dysfunction leads to the accumulation of damaged mitochondria, increases in ROS levels, and deficiency of cellular energy, which, in the brain, induce synaptic dysfunction, cognitive deficits, and compromising neuroplasticity [183,184].

Autophagy is a process that allows maintaining metabolic homeostasis thanks to its sensitivity to nutrients presence. Under hunger or calorie restriction conditions, autophagy is activated to provide a readily usable substrate for energy production [185]. In contrast, when nutrients are abundant, autophagy is attenuated [185,186]. 

Insulin inhibits autophagy by activating the PI3K/mTOR pathway, which phosphorylates and inhibits the ULK1 complex responsible for initiating phagophore formation in mammals [187]. Likewise, insulin can induce, via the IRS/PI3K/Akt/FoxO pathway, the activation of ATG gene expression [186,188]. Under amino acids and nutrient starvation, mTORC1 regulates autophagy through inhibition of ULK and dephosphorylation of protein phosphatase 2A (PP2A), which activates the class III phosphatidylinositol 3-kinase complex. This complex contains Beclin 1, VPS34, VPS15, and ATG14, which induce phagophore formation and activate autophagy [189,190]. 

The regulation of mTOR and FoxO transcription factors by the insulin/IRS/Akt pathway has key implications for mitochondrial function since mTOR1 [191,192] and FoxO1 control mitochondrial biogenesis oxidative function through the regulation of PGC-1α [193,194].

Recently, the correlation between mitophagy and diabetes is emerging. Mitophagy contrasts the degenerative processes that arise from muscle insulin resistance and β cells dysfunction by interrupting a series of events leading to oxidative stress and mitochondrial damage. 

Sidarala et al. observed that, in diabetic cellular models, ROS induced by proinflammatory cytokines caused mitochondrial damage and activation of mitophagy. When mitophagy was compromised, an increase in β-cell death and hyperglycemia was detected, concluding that mitophagy prevents diabetes by promoting β cell survival and countering inflammatory injury in both human and rodent β cells [195]. 

Using mice with specific beta cell Atg7 deletion, Jung et al. [196]. showed that autophagy deficiency causes apoptosis in beta cell and decreased pancreatic insulin content, glucose tolerance and serum insulin level. Thus, these results confirmed the important role of autophagy for the structural and functional integrity of pancreatic beta cells and its role in diabetes development.

The PINK1/Parkin pathway has received more attention in recent years due to an abnormal increase in autophagic vacuoles rich in defective mitochondria.

In diabetic retina of db/db mouse was observed a modulation of PINK and Parkin proteins with reduction in mitophagic autophagosome number and increase in oxidative stress. In addition, in T2D patients a reduction in genic expression of PINK and Parkin was observed [197]. 

Further, abnormalities in PINK and Parkin’s activity were found in neurodegeneration associated with diabetes. For example, in the brain cortex of 6-month-old Goto–Kakizaki (GK) rats, an animal model of non-obese type 2 diabetes, a reduction in the LC3-II protein levels and a concomitant increase in mTOR phosphorylation at Ser-2448 levels suggested the suppression of autophagy [198]. Significant Parkin loss and PGC-1α reduction were found in the substantia nigra (SN) of db/db mice and high-fat-diet (HFD) mice. Interestingly, these alterations were reversed by the administration of metformin, a well-known antidiabetic drug [198]. In addition, increased Parkin translocation into depolarized mitochondria and increased autophagic flux occur in neurons overexpressing APP (hAPP) and in the brains of patients with AD. Thus, autophagy is compromised, resulting in defective mitochondria accumulation [184]. Similarly, a significant reduction in Beclin 1 expression and the autophagic process was observed in AD patients [199]. 

It has been proposed that Aβ and Tau, implicated in AD pathology and other tauopathies, can interfere with the mitophagic process by sequestering cytosolic Parkin, which can no longer ubiquitinate dysfunctional mitochondria interfering with MQC [200]. In AD mice engineered for a single mutation in the Beclin/BECN1 gene, the binding of BECN1 with the BCL2 inhibitory protein was reduced. Therefore, this led to constitutively active autophagy, which sequestered amyloid oligomers in both the cortex and hippocampus and prevented the progression of AD [201]. 

These outcomes were also confirmed by studies on Atg7 Knockout mice showing that autophagy deficiency significantly reduced the presence of extracellular amyloid plaque in excitatory neurons of the forebrain. The authors justified this reduction by explaining that it was due to inhibition of Aβ secretion, which led to intraneuronal Aβ accumulation in the perinuclear region [202]. Interestingly, after rapamycin treatment, 3xTg-AD mice recovered cognitive functions following activation of mTOR-mediated autophagy and reduction in Aβ levels [203]. This review collects the latest works that correlate mitophagy with AD pathology [204].

A defect in the fusion process between autophagosomes and lysosomes can also be responsible for lack of mitophagy. In CA1 neurons of the AD brain, autophagy is progressively upregulated, despite an impairment in the ability to eliminate autophagic substrates [205]. A deficit in lysosomal acidification was found in cellular and animal models, probably caused by presenilin 1 mutations [206]. 

The mitochondrial function, regulated by mitophagy, is essential to preserve the pancreatic beta cells’ activity and insulin secretion, and an alteration of mitophagic clearance exacerbates the IR induced by a high-fat diet [207]. In this condition, defective mitophagy leads to the accumulation of damaged mitochondria and the production of large amounts of ROS, which activate the NLRP3-ASC-caspase1 pathway, inducing insulin resistance and cognitive decline [201,207,208].

Some data have suggested a direct relationship between autophagic processes and the development of T2D. For instance, diabetic mice db/db and C57BL/6 HFD mice showed activation of autophagy and upregulation of autophagosome formation in beta cells in the presence of insulin resistance [207]. Genetic ablation of Atg7 in beta cells resulted in islet degeneration and impaired glucose tolerance with reduced insulin secretion [209]. Caccamo et al. [203] investigated the status of the PI3K/Akt/mTOR pathway in post-mortem tissue from the inferior parietal lobule (IPL) at three different stages of AD (pre-clinical AD (PCAD), mild cognitive impairment (MCI) and late-stage AD). In the early stages of AD, it was found hyperphosphorylation of Akt, PI3K (p85 subunit), and hyperactivation of mTOR was associated with the reduction in autophagy [203]. Further, a significant increase in Aβ_1-42_ levels coupled with the decrease in autophagy and the progression of the disease was found. An increase in insulin receptor substrate 1, a candidate biomarker of brain insulin resistance, was found in MCI and AD patients, providing an additional link between AD pathology and insulin resistance [206]. Other studies in animal models of obesity and T2D reported a downregulation of PINK associated with mitophagy impairment, showing a link between mitochondrial quality control and insulin sensitivity [142,210].

These data led to the assumption that the impairment of autophagy could be an early event in diseases related to insulin resistance and that mitochondrial dysfunction plays an essential role in the onset and progression of these diseases. Moreover, pharmacological or nutritional interventions that reestablish mitophagy homeostasis and help the elimination of damaged mitochondria could act as a potential therapy in metabolic and neurodegenerative diseases. 

## 15. Mitochondrial Biogenesis

Mitochondrial biogenesis (mitobiogenesis) is a process that, through delicate coordination of nuclear and mitochondrial events, leads to the growth and division of pre-existing mitochondria in response to energy demand, triggered by stressors or developmental signals. The complexity of this process goes beyond the scope of this review, and this topic has been well developed by Gureev et al. [211]. Here, we will briefly summarize the major transcriptional regulators.

Peroxisome-proliferator-activated receptor-gamma coactivator (PGC)-1alpha (PGC-1α) is considered the main regulator of mitochondrial biogenesis (Figure 2) [212]. It is mainly expressed in tissues rich in mitochondria and with a high energy requirement, such as the brain, heart, skeletal muscles, brown adipose tissue, and kidneys [213,214]. Its transcription increases when greater cellular ATP demand, including exercise, cold and fasting, is required [215]. Upon activation, PGC-1α controls the expression of a number of transcription factors, such as respiratory nuclear factor-1 and -2 (NRF-1 and NRF-2), peroxisome-proliferator-activated receptors (PPARs) [216], and the estrogen-related receptor (ERR)-α and γ [217] which regulate the expression of mitochondrial proteins encoded by nuclear DNA (nDNA), such as TOMM20 and electron transfer chain (ETC) subunits [218,219]. Furthermore, it promotes the expression of mitochondrial transcription factor A (TFAM) [220], which, by regulating the expression of the 13 mitochondrial genes (tRNA, rRNA, and the subunits of the respiratory chain), drives mtDNA transcription and replication [211,221,222]. Thereby, it coordinates mitochondrial biogenesis between the two genomes. Furthermore, TFAM, thanks to its ability to compact mitochondrial DNA into nucleoids, protects it from oxidative damage [223]. 

After transcription and translation, nDNA-encoded mitochondrial proteins are folded in a process coupled to the outer membrane translocase (TOM), transported to the mitochondrial matrix, assembled, and sorted into a precise intramitochondrial location [224].

Given the important role in the bioenergetic response, the activity of PGC-1α has finely regulated through phosphorylation and acetylation/deacetylation processes by employing various metabolic sensors. In this review, we will pay attention to AMPK and SIRT1 proteins involved in metabolic and bioenergetic functions and that result to be attractive targets for the treatment of neuronal dysfunctions associated with dysmetabolism. Interestingly, AMPK and SIRT1 are also implicated in mitophagic clearance beyond regulating mitochondrial biogenesis. 

The transcription and activity of PGC-1α are enhanced by SIRT1 through deacetylation to specific lysine residues [225], or by AMPK via phosphorylation at Thr-177 and Ser-538 [225,226]. In addition, p38 MAPK increases the activity of PGC-1α by direct phosphorylation at Thr-262, Ser-265, and Thr-268, leading to the inhibition of PGC-1α and its repressor p160^MBP^ interaction [227,228].

Furthermore, PGC-1α is negatively regulated both by phosphorylated Gsk3β, which, in response to acute oxidative stress, activates its intranuclear proteasomal degradation [229], and by acetylation due to general control of amino acid synthesis 5 (GCN5) acetyltransferase [230]. Insulin acts as a negative regulator of mitochondrial biogenesis by inhibiting the activity of PGC-1α through Akt. It can directly act by phosphorylating the Ser-570 residue on PGC-1α, or indirectly through the phosphorylation of Clk2 kinase [230]. Furthermore, the insulin/IRS/PI3K/Akt pathway controls mitochondrial biogenesis and oxidative function by regulating PGC-1α through mTOR [191,192] and FoxO, whose cascade suppresses the expression of PGC1α [136,193,194].

## 16. AMPK

AMP-activated protein kinase (AMPK) is the main factor in the post-translational modification of PGC-1α. AMPK is a heterotrimeric complex that acts as a cellular energy sensor capable of regulating mitochondrial biogenesis. It is composed of a catalytic subunit α, which activates AMPK after phosphorylation at Thr172 by the calcium-sensitive calmodulin-dependent protein kinase-beta (CaMKKβ) [231] and by the LKB1 complex (LKB1/STRAD/MO25) [232,233]. It also possesses two other regulatory subunits β and γ, which allow the functioning of AMPK as a glycogen and energy sensor [234].

AMPK is activated in response to stressful conditions, such as low glucose, hypoxia, exercise, and ischemia, to restore physiological ATP levels by inhibiting energy-consuming anabolic processes (e.g., gluconeogenesis, fatty acid, protein synthesis, and cell growth) [235,236]. It works through the inhibition of the mTOR 1 complex, increasing the energy production through catabolic reactions, including glucose uptake and glycolysis, fatty acid oxidation, and mitochondrial biogenesis [237,238]. Hyperinsulinemia, accompanied by excessive nutrient accumulation, inhibits AMPK through inhibitory phosphorylation on S^485/491^ mediated by Akt [239]. 

## 17. SIRT1

SIRT1 belongs to the class III histone deacetylases family [240]. Seven mammalian sirtuins are known: three (Sirt3, Sirt4, and Sirt5) are called mitochondrial sirtuins [241]. Although SIRT1 is mainly nuclear, it acts on mitochondrial biogenesis and turnover [240]. Its activity depends on the coenzyme nicotinamide adenine dinucleotide (NAD+) and is highly sensitive to changes in the cells’ energy demand. Under conditions of fasting, exercise, or oxidative stress, it increases PGC-1α deacetylation to specific lysine residues [210,230,242], resulting in increased transcription of PGC-1α target genes and activation of mitochondrial biogenesis [242,243].

The role of SIRT1 as a metabolic regulator is enhanced by its ability to interact with the other two metabolic sensors. It is, in fact, able to activate AMPK by employing hepatic kinase B1 (LKB1) [244,245].

Furthermore, SIRT1, by interacting with TSC2, a component of the mTOR complex upstream to mTORC1, negatively regulates mTOR signaling [246]. SIRT1, besides its role in regulating mitochondrial biogenesis, is also involved in the turnover of defective mitochondria by mitophagy [240]. 

## 18. Mitochondrial Biogenesis and Insulin

In order to guarantee the ATP necessary for physiological cellular processes, mitochondria are able to respond to different metabolic conditions by promoting crosstalk with the nucleus to induce the expression of specific genes. In particular, this mito-nuclear communication promotes the expression of nuclear and mitochondrial genes involved in mitochondrial function, including mitobiogenesis, oxidative metabolism, and the mitochondrial stress response in conditions of dysmetabolism, such as in T2D and IR, are downregulated [10,247,248]. 

Several human and rodent studies have demonstrated that neuronal insulin/IGF-1 signaling can modify the mitochondrial function, directly affecting mitochondrial biogenesis and oxidative capacity [10,249]. The insulin deficiency occurring in subjects with IR or in rats after high-fat feeding reduces the production of muscle mitochondrial ATP and the expression of oxidative phosphorylation genes. Conversely, insulin infusion leads to an enhanced expression of mitochondrial proteins, higher oxidative enzyme activity, and an increased synthesis of ATP in the muscles [133].

As previously mentioned, PGC-1α is considered the main controller of mitochondrial biogenesis and metabolism and contributes to the alterations in mitochondrial number and function observed in various metabolic and neurodegenerative diseases [250]. The expression of PGC-1α and most of its target genes is, in fact, significantly reduced in insulin-resistant and diabetic subjects [217,251], as well as in the brains of patients with neurodegenerative conditions [251]. In agreement, some studies have shown that the activation of SIRT1 and PGC-1α improves mitochondrial function and insulin sensitivity [252].

Moreover, several groups report a correlation between some human genetic variants of PGC-1α and the onset of diabetes and insulin resistance [253]. PGC-1α regulates mitochondrial density in neurons and counteracts oxidative stress by controlling the expression of genes related to the generation of reactive oxygen species (ROS) [254]. PGC-1α silencing inhibits synaptogenesis and reduces the dendritic spine density in in vivo hippocampal dentate granule neurons [255]. In AD pathology, altered mitobiogenesis induced by impaired PGC-1α activity triggers degeneration of neurons, inducing mitochondrial and synapse dysfunction, and contributes to cognitive decline [256]. 

Further, studies in mice have demonstrated that, depending on the tissue, an increased expression of PGC-1α may be beneficial or harmful [257]. In adipose tissue, it promotes thermogenesis, protecting against energy overload arising in diabetes and obesity. Similarly, in muscle, PGC-1α stimulates a modification of the phenotype regarding oxidative metabolism. In contrast, a dangerous role is played in the liver and pancreas, where it stimulates hepatic gluconeogenesis and inhibits insulin secretion in response to glucose, stimulating diabetes onset [257]. Furthermore, the pancreas of obese and diabetic rodents showed high levels of PGC-1α mRNA, suggesting that PGC-1α dysregulation may be involved in these disorders and, consequently, might be associated with neurodegeneration [257].

As already exposed in this review, intranasal insulin administration recuperates memory and learning and increases the ATP levels in the brain [117,123]. Further, in rodent models, intranasal insulin improves mitochondrial biogenesis, increasing PGC1α expression and mitochondrial DNA content in the brain hippocampus region. Thus, intranasal insulin acts on mitochondrial function and biogenesis, leading to new evidence that it could have therapeutic benefits for dementia associated with metabolic disease [143].

## 19. Mitochondrial Therapy: Targeting Mitochondria to Break down the Damage

Although the mechanism underlying the onset of insulin resistance and the associated neuronal death is not yet clear several reports, as previously described, indicate mitochondrial dysfunction as one of the protagonists of both events. Thus, targeting the mitochondria could be a winning move in addressing these pathologies. Effective mitotherapy should ensure the maintenance of an efficient mitochondria pool able to cope with stressful situations, guarantee the production of ATP, and maintain homeostasis and cellular viability. Therefore, the most effective strategies should aim to generate new mitochondria (by acting on biogenesis) or at the removal or replacement of dysfunctional mitochondria (through mitophagy or transplantation of healthy mitochondria), or finally, they could act on the consequences of mitochondrial dysfunction. 

“Mitoceuticals” is a term coined to indicate drugs having a target mitochondrial function. Some drugs are rapidly acquiring a great interest in developing new directed treatments.

For example, imeglimin is the first of a new class of oral antidiabetic agents containing tetrahydrotriazine called “glimins”, which works by correcting the mitochondrial dysfunction of various organs involved in type 2 diabetes. It, indeed, re-establishes the activity of the respiratory chain with a consequent reduction in ROS production and prevention of the mitochondrial permeability transition pores opening. So far, the compound has successfully passed phase I and II clinical trials in multiple countries to evaluate the efficacy and tolerability in T2D patients, both as a monotherapy and as an adjunct therapy in patients ineffectively controlled by hypoglycemic drugs such as metformin. In June 2021, imeglimin received its first approval for use in T2D in Japan [258].

Drugs such as MitoQ, based on coenzyme Q10 [259], or SkQ1, a derivate of plastoquinone [260], having a high antioxidant effect, are specifically targeted to mitochondria and can be administered as dietary supplements [261]. Equally, an increased interest concerns studies on pharmacological induction of mitochondrial biogenesis. Molecules such as thiazolidinediones (TZDs) or bezafibrate are agonists of peroxisomal-proliferator-activated receptors (PPARs) and their downstream target PGC-1α [262]. Pioglitazone, an FDA-approved drug with insulin-sensitizing and lipid-lowering effects, is one of the most promising TZDs for the treatment of mitochondrial dysfunction. This drug is capable to promote biogenesis and mitochondrial function in diabetic patients. 

The analogues MSDC-0602K and MSDC-0160 are currently submitted in several clinical trials. MSDC-0160 (mitoglitazone) showed improved brain glucose metabolism and decreased brain damage in AD patients [263]. A phase III clinical trial (ClinicalTrials.gov: NCT03970031) will soon begin to evaluate glycemic control and cardiovascular outcomes in subjects with pre-T2D or T2D and evidence of NAFLD/NASH of MSDC-0602K.

Dysfunctional mitophagy is a common event in neurodegenerative and dysmetabolic disorders, and the development of molecules capable of restoring mitophagy can, therefore, represent a promising therapeutic strategy. 

Several inducers of mitophagy have been tested in various in vivo and in vitro studies, and some of them have been part of larger clinical studies. The precursors of NAD^+^, nicotinamide mononucleotide or nicotinamide (NAM) or energy modulators, including resveratrol, were demonstrated to be able to promote mitophagy through AMPK and SIRT1 signaling [262]. Modulation of these molecules reduced Aβ_1-42_ levels and improved cognitive impairment in different AD models [264,265]. Several direct and indirect AMPK activators have been primarily developed for the treatment of metabolic diseases, such as diabetes and non-alcoholic liver conditions. Among them, PXL770, a direct oral AMPK activator, has successfully completed phase I clinical trials to evaluate safety and tolerability in diabetic subjects. Further, the results of one phase II study shows encouraging data for the treatment of NASH and T2D comorbidity compared to placebo [266]. 

However, classic mitochondrial therapies are not yet able to provide lasting and decisive protection for many pathologies. In recent years, great attention and hopes have been focused on mitochondrial transplantation. It represents a promising strategy aimed at the replacement of damaged mitochondria with autologous or heterologous healthy mitochondria to restore homeostasis, survival, and neuronal regeneration [267]. Several works support its effectiveness in various neurodegenerative diseases, such as stroke, spinal cord injury, and Parkinson’s disease [267,268]. McCulley et al. developed mitochondrial transplantation as a therapeutic approach to treat ischemia in pediatric patients [269]. Mitochondria, isolated from the patient’s own body, were injected into the ischemic hearth, and the function of native damaged mitochondria and the cellular viability was restored. The authors conclude that mitochondrial transfer can be applied for the treatments of other injuries in which the organelle is damaged [269]. Various techniques have been used for the transfer of exogenous mitochondria, generally extracted from peripheral muscle tissues, into the CNS, including direct intracerebral microinjection [270], unfortunately considered highly invasive, or intravenous and intranasal administration studied in PD [271,272]. 

Each of these routes of administration shows limits that reduce the effectiveness of the treatment, such as the reduced number of transplanted mitochondria able to cross the BBB, reach the lesion site, and pass the neuronal cell membrane [273]. Recently, the efforts of several groups have aimed at improving the quality of mitochondrial transfer to the CNS. In particular, the efforts were focused on increasing the efficacy of mitochondrial delivery and improving the internalization of exogenous mitochondria. For this last aim, foreign mitochondria have been conjugated with a cell penetrating peptide, Pep-1, and an increase in the cellular uptake and rescue of mitochondrial function in cybrid cells, used as a model of mitochondriopathies, was demonstrated [274]. In order to enhance cellular transplantation, functionalization of the mitochondrial surface with biocompatible hydrophilic polymers, such as dextran, was developed, and the efficiency of transplantation with respect to naked mitochondria was validated both in in vitro and in vivo systems [275]. Finally, in a recent paper, Picone et al., identified synaptosomes, vesicles obtained from synaptic terminations, as a possible natural vehicle to deliver healthy mitochondria directly into the cytoplasm of neuronal target cells [276]. Here, they were able to rescue the mitochondrial function of human neuroblastoma cells damaged by treatment with rotenone or m-chlorophenyl hydrazone carbonyl cyanide (CCCP) [276].

Mitochondrial therapy is still in an initial (experimental) phase despite the excellent premises. Further studies are needed to better understand the mechanisms of mitochondrial delivery and internalization in target cells and investigate the possible immunological response following transplantation to improve the conditions of cell–cell mitochondrial transfer concerning the different damage sites.

## 20. Conclusions

Several studies have demonstrated that altered insulin signaling in metabolic dysfunction and its comorbidities is not only a disorder of peripheral tissues but also comprises brain dysregulation and altered metabolic physiology. Alterations in the mitochondrial structure and function can lead to cell damage, culminating in neuronal loss. Disruption of MQC mechanisms, mainly the balance between biogenesis and mitophagy, can trigger degenerative processes. Finally, understanding the mechanism underlying the mitochondrial dysfunction, and perhaps the crosstalk with other organelles, particularly lysosomes and ER, may provide therapeutic potential for dysmetabolism and neurodegeneration. Further, targeting drugs to mitochondria represents a new trend in molecular pharmacology for treating the neurodegenerative diseases associated with metabolic disorders. Finally, the possibility of transplanting fully functional mitochondria directly into defective cells represents an appealing strategy to cure pathologies related to mitochondrial dysfunction.

## Figures and Tables

**Figure 1 biology-11-00943-f001:**
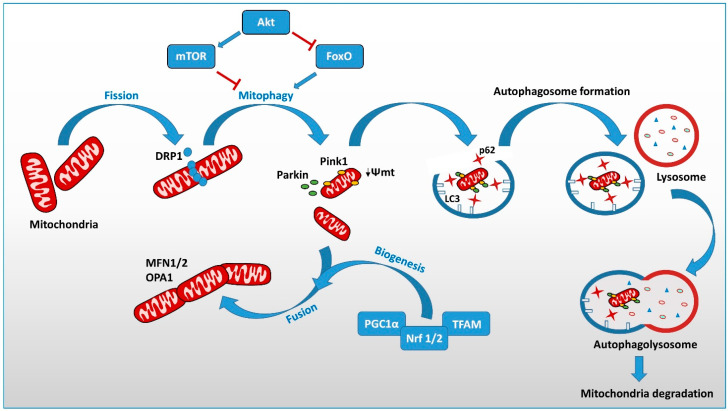
Schematic representation of mitochondrial dynamics and mitophagy process.

**Figure 2 biology-11-00943-f002:**
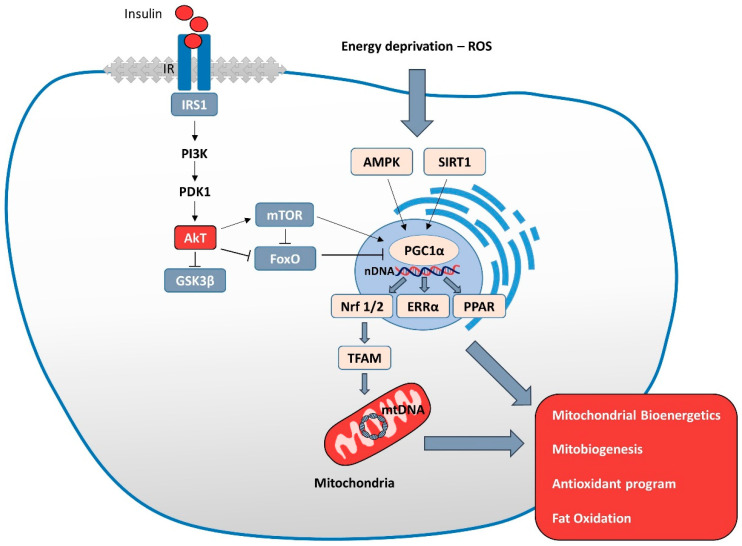
Schematic representation of PGC-1α-driven biogenesis and mitochondrial regulation and its interactions with some of its target transcription factors involved in metabolic regulation in response to nutrient availability or energy demand. The detailed action of each regulatory path is discussed in the text.

## Data Availability

Not applicable.

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
