# Peer review of "Insulin and Its Key Role for Mitochondrial Function/Dysfunction and Quality Control: A Shared Link between Dysmetabolism and Neurodegeneration"

_biology, 2022, doi:10.3390/biology11060943_

Round 1

Reviewer 1 Report

In the present form, your review is very misleading. You should pay more attention to the brain energy metabolism and physiology. Please note that insulin is used by peripheral tissues that accumulated glucose, especially skeletal muscles, the lives, and fat tissue. Please note that in physiological conditions, insulin cannot trespass the blood-brain barrier. In the brain, you should rather talk about the activity of the IGF. That is why the results of insulin impact on activity in the brain slices are worthless. The same problem is with neurodegenerative diseases and Alzheimer's disease in particular. The massive neuronal cell dying in AD is rather an effect of deficient glucose metabolism. You should read the review: "Energy Metabolism Decline in the Aging Brain—Pathogenesis of Neurodegenerative Disorders". Metabolites 10 (450), doi:10.3390/metabo10110450

Author Response

Referee n° 1

We would like to thank the reviewer for his valuable comments which further have helped us to improve the quality of our manuscript. 

In the present form, your review is very misleading. You should pay more attention to the brain energy metabolism and physiology. Please note that insulin is used by peripheral tissues that accumulated glucose, especially skeletal muscles, the lives, and fat tissue. Please note that in physiological conditions, insulin cannot trespass the blood-brain barrier.

According to the referee’s comment, a sentence regarding the use of insulin by peripheral tissues that accumulates glucose such as skeletal muscles, liver, and fat tissue has been added in section 3 [Ref.19], and the active transport across the BBB is discussed in section 6.

In the brain, you should rather talk about the activity of the IGF.

We would like to thank the referee for having underlined this point and a sentence about IGF-1 protective activity in the brain has been included in section 5. However, IGF activity was already partially described in sections 10 and 18.

The same problem is with neurodegenerative diseases and Alzheimer's disease in particular. The massive neuronal cell dying in AD is rather an effect of deficient glucose metabolism.

According to the referee’s suggestion, a sentence regarding the role of glucose metabolism in neurodegeneration has been included in section 10.

You should read the review: "Energy Metabolism Decline in the Aging Brain—Pathogenesis of Neurodegenerative Disorders". Metabolites 10 (450), doi:10.3390/metabo10110450

The authors thank the reviewer for suggesting the reading of this paper and the content has been integrated into section 10 and the reference section  [Ref.130]

Reviewer 2 Report

The review summarize important knowledge regarding insulin and mitochondria and their role in dysmetabolism and neurodegeneration, the two highly increasing problems of today world. Thus the review is of most relevance.

I see only few formal issue to be addressed:

lines 97-99 - Sentence is hard to understand. Can you reformulate or split into two sentences.

lines 207-209 - Sentence is hard to understand. Can you reformulate or split into two sentences.

There is issue with font size on several places: lines 222-224, 311, 369-372

line 233 - there seems to be missing word behind "where'

in Part 8, lines 383-448, there is missing beta behind A on several places

Author Response

We would like to thank the reviewer for his positive evaluation and  valuable comments which further have helped us to improve the quality of our manuscript.

The review summarize important knowledge regarding insulin and mitochondria and their role in dysmetabolism and neurodegeneration, the two highly increasing problems of

I see only a few formal issues to be addressed:

lines 97-99 - Sentence is hard to understand. Can you reformulate or split into two sentences?

As suggested by the referee n°4 section 1 has been summarized and the sentence eliminated

lines 207-209 - Sentence is hard to understand. Can you reformulate or split into two sentences?

The sentence has been reformulated

There is issue with font size on several places: lines 222-224, 311, 369-372

The font size on lines 222-224, 311, 369-372 in the original file is correct, it is probably an editorial formatting problem

line 233 - there seems to be missing word behind "where'.

According to the referee’s comment, the sentence has been rewritten.

in Part 8, lines 383-448, there is missing beta behind A on several places

Greek letters (beta) are present in the original manuscript, (see also Gsk3 but are missing in the editorial file, it is probably an editorial formatting problem

Reviewer 3 Report

The review by Galizzi and Di Carlo's entitled “Insulin and its key role for mitochondrial dys/function quality control. A shared link between dysmetabolism and neurodegeneration” presents an overview of insulin and related topics, ranging from insulin history to structure, synthesis and effects on targets such as specific organs and tissues, under physiological and pathological conditions. Several interesting topics are covered, such as the effect of metabolic disease and insulin resistance in the brain and insulin signaling in mitochondrial homeostasis. Furthermore, a specific focus on mitochondrial biogenesis and quality control is also provided. Finally, the authors present updates on the potential of mitochondrial therapy as a promising approach to address metabolic and neurodegenerative diseases.  

The review is dense of information, concepts are clearly presented and broadly discussed; in some cases, I found a bit of redundancy. Overall, the manuscript needs some improvements. 

Major point

Lines 162-164. “Insulin secretion may be induced by variations in synthesis at the level of gene transcription, translation, and post-translational modification in the Golgi and factors mediating insulin release from secretory granules”. Since the sentence was unclear, I checked the reference and I found that your sentence is too close to the source (“Insulin secretion may be influenced by alterations in synthesis at the level of gene transcription, translation, and post-translational modification in the Golgi as well as by factors influencing insulin release from secretory granules”. Wilcox, G; Insulin and Insulin Resistance Clin Biochem Rev May 2005, 26, 19-39). Therefore, the authors are kindly asked to clarify the concepts by rewriting the passage; alternatively, they should delete the sentence. Finally, I hope there are no other cases like this elsewhere in the manuscript.

Minor points

Here are some examples of language deficiencies, syntax errors, leaving the authors to find and correct further inaccuracies (there are many). 

In the title  “dys/function” is “dysfunction”. 

Line 58: "who" is "what"

Line 182:" Engeenrized" (?)  is "engineered", I guess

Paragraph 8: Alzheimer disease and insulin. The authors should mention aducanumab as the first drug recently approved in the United States to treat some cases of Alzheimer disease. 

I would strongly suggest to avoid, especially when related to biochemistry, the terms “reduce, reduction” with the meaning of “decrease”. This could avoid oxymoron like “the reduced oxidation” (line 580). 

Line 820: if FIS1 is the same as in line 673, it should be capitalized or vice versa

I suggest paying attention to the placement of commas, as I have found extensive use of them, especially in front of "and" when not required. Similarly, for the use of “and”.

Some sentences are too long (i.e. lines 761-764; 597-601), and should be rephrased to improve clarity.

Paragraph numbering is correct till n° 17 (SIRT1), afterwards it restarts with n° 15. Please, amend.

Author Response

We would like to thank the reviewer for his valuable comments which further have helped us to improve the quality of our manuscript.

Major point

Lines 162-164. “Insulin secretion may be induced by variations in synthesis at the level of gene transcription, translation, and post-translational modification in the Golgi and factors mediating insulin release from secretory granules”. Since the sentence was unclear………

Thank you for having evidenced this oversight the sentence has been rewritten

Minor points

Here are some examples of language deficiencies, syntax errors, leaving the authors to find and correct further inaccuracies (there are many). 

In the title “dys/function” is “dysfunction”. 

Line 58: "who" is "what"

Line 182:" Engeenrized" (?)  is "engineered", I guess

I would strongly suggest to avoid, especially when related to biochemistry, the terms “reduce, reduction” with the meaning of “decrease”. This could avoid oxymoron like “the reduced oxidation” (line 580). 

Line 820: if FIS1 is the same as in line 673, it should be capitalized or vice versa

I suggest paying attention to the placement of commas, as I have found extensive use of them, especially in front of "and" when not required. Similarly, for the use of “and”.

Some sentences are too long (i.e. lines 761-764; 597-601), and should be rephrased to improve clarity. 

Paragraph numbering is correct till n° 17 (SIRT1), afterwards it restarts with n° 15. Please, amend.

In response to all these comments, the test has been corrected.

Paragraph 8: Alzheimer disease and insulin. The authors should mention aducanumab as the first drug recently approved in the United States to treat some cases of Alzheimer disease. 

We would like to thanks the referee for having suggested the inclusion of this drug in the test. This information and related reference have been now integrated in section 8 [Ref.100]

Reviewer 4 Report

This review paper discusses the interplay between mitochondrial dysfunction/quality control and insulin in the context of neurodegenerative diseases. This manuscript is well written, and most of the chapters shed significant light on the mechanisms of insulin secretion for glucose metabolism, mitochondrial dynamics, and its quality control. I have read the manuscript and have a few suggestions to the manuscript.

Major Comments:

  • The sections and the information presented in this manuscript is adequately presented. However, I believe the authors can probably do away with the section of ‘Insulin Tales” or probably be more succinct in this section. I believe this section does not add too much to the overall goal of this paper.

  • Section 14: “Mitophagy and Insulin” – This section covers some aspects of the interplay between Mitophagy and Insulin, however, the information does not seem adequate, and the section feels more like a continuation from section 13 i.e. “Mitophagy”. It would be great if the authors would focus more on the link between mitophagy and insulin by highlighting studies that have been conducted in this area.

  • Section 16: “Mitochondrial therapy: targeting mitochondria to break down damage” – This section outlines therapeutic strategies for targeting the mitochondria, however, it does not discuss these strategies in terms of insulin resistance or diabetes. A quick google search on this topic shows studies in mice where the mitochondria are targeted to alleviate diabetes or demonstrate some therapeutic intervention. It would be great if the authors would bring this correlation in their discussion rather than just discuss drugs that target the mitochondria in general which seems more apt for mitochondrial disease treatment/management.

Author Response

We would like to thank the reviewer for his positive evaluation and  valuable comments which further have helped us to improve the quality of our manuscript.

This review paper discusses the interplay between mitochondrial dysfunction/quality control and insulin in the context of neurodegenerative diseases. This manuscript is well written, and most of the chapters shed significant light on the mechanisms of insulin secretion for glucose metabolism, mitochondrial dynamics, and its quality control. I have read the manuscript and have a few suggestions to the manuscript.

We would like to thank the reviewer for his positive evaluation.

Major Comments

The sections and the information presented in this manuscript is adequately presented. However, I believe the authors can probably do away with the section of ‘Insulin Tales” or probably be more succinct in this section. I believe this section does not add too much to the overall goal of this paper.

The section Insulin Tales has been summarized.

Section 14: “Mitophagy and Insulin” – This section covers some aspects of the interplay between Mitophagy and Insulin, however, the information does not seem adequate, and the section feels more like a continuation from section 13 i.e. “Mitophagy”. It would be great if the authors would focus more on the link between mitophagy and insulin by highlighting studies that have been conducted in this area.

The section has been reviewed and new studies have been mentioned in section 14 [Ref. 195,196,197.

Section 16: “Mitochondrial therapy: targeting mitochondria to break down damage” – This section outlines therapeutic strategies for targeting the mitochondria, however, it does not discuss these strategies in terms of insulin resistance or diabetes. A quick google search on this topic shows studies in mice where the mitochondria are targeted to alleviate diabetes or demonstrate some therapeutic intervention. It would be great if the authors would bring this correlation in their discussion rather than just discuss drugs that target the mitochondria in general which seems more apt for mitochondrial disease treatment/management.

As suggested by the reviewer in section 19 we have included new appropriate studies  and integrated in the Reference section [Ref. 258,263,266]